# A pilot trial to evaluate the clinical usefulness of contrast-enhanced ultrasound in predicting renal outcomes in patients with acute kidney injury

Hye Eun Yoon[1], Da Won Kim[1], Dongryul Kim[1], Yaeni Kim[2], Seok Joon Shin[1], Yu Ri Shin[3]*

1 Department of Internal Medicine, Incheon St. Mary's Hospital, College of Medicine, The Catholic University of Korea, Incheon, Republic of Korea, 2 Department of Internal Medicine, Seoul St. Mary's Hospital, College of Medicine, The Catholic University of Korea College of Medicine, Seoul, Republic of Korea, 3 Department of Radiology, Seoul St. Mary's Hospital, College of Medicine, The Catholic University of Korea, Seoul, Republic of Korea

* crystal57@daum.net

**Data Availability Statement:** All relevant data are within the paper and its Supporting Information files.

## Abstract

### Objectives

Contrast-enhanced ultrasound (CEUS) enables the assessment of real-time renal microcirculation. This study investigated CEUS-driven parameters as hemodynamic predictors for renal outcomes in patients with acute kidney injury (AKI).

### Methods

Forty-eight patients who were diagnosed with AKI were prospectively enrolled and underwent CEUS at the occurrence of AKI. Parameters measured were the wash-in slope (WIS), time to peak intensity, peak intensity (PI), area under the time–intensity curve (AUC), mean transit time (MTT), time for full width at half maximum, and rise time (RT). The predictive performance of the CEUS-driven parameters for Kidney Disease Improving Global Outcomes (KDIGO) AKI stage, initiation of renal replacement therapy (RRT), AKI recovery, and chronic kidney disease (CKD) progression was assessed. Receiver operating characteristic (ROC) analysis was performed to evaluate the diagnostic performance of CEUS.

### Results

Cortical RT (Odds ratio [OR] = 1.21) predicted the KDIGO stage 3 AKI. Cortical MTT (OR = 1.07) and RT (OR = 1.20) predicted the initiation of RRT. Cortical WIS (OR = 76.23) and medullary PI (OR = 1.25) predicted AKI recovery. Medullary PI (OR = 0.78) and AUC (OR = 1.00) predicted CKD progression. The areas under the ROC curves showed reasonable performance for predicting the initiation of RRT and AKI recovery. The sensitivity and specificity of the quantitative CEUS parameters were 60–83% and 62–77%, respectively, with an area under the curve of 0.69–0.75.

**Funding:** Dr. Yu Ri Shin is supported by the Catholic Medical Center Research Foundation made in the program year of 2017. The funders had no role in study design, data collection and analysis, decision to publish, or preparation of the manuscript.

**Competing interests:** The authors have declared that no competing interests exist.

**Abbreviations:** AKI, acute kidney injury; AUC, area under the time-intensity curve; CEUS, contrast-enhanced ultrasound; CKD, chronic kidney disease; eGFR, estimated glomerular filtration rate; FENa, fractional excretion of sodium; FWHM, time for full width at half maximum; KDIGO, Kidney Disease Improving Global Outcomes; MTT, mean transit time; PI, peak intensity; ROI, region of interest; RRT, renal replacement therapy; RT, rise time; TIC, time–intensity curve; TTP, time to peak intensity; WIS, wash-in slope.

## Conclusion

CEUS may be a supplemental tool in diagnosing the severity of AKI and predicting renal prognosis in patients with AKI.

## Introduction

Acute kidney injury (AKI) is characterized by a rapid decline in kidney function within a few hours to a few days. AKI is responsible for two million deaths annually worldwide, and its incidence is increasing [1]. Although alterations in renal perfusion are thought to play a central role in its pathogenesis [2], diagnostic tools for assessing renal perfusion are lacking.

Different imaging modalities, such as computed tomography (CT), magnetic resonance imaging (MRI), and positron emission tomography, are limited in clinical applications due to their high cost, reduced availability, long examination duration, and toxicities associated with the contrast agents used. Kidney ultrasound (US) is the most widely used imaging modality in the initial workup of AKI, because it is widely available and free of complications [3]. The rate of abnormal US findings in cases of AKI is not high, because different renal parenchymal diseases often display the same US appearance, whereas the same renal parenchymal disease may present different appearances on US according to the disease stage [4]. Doppler US provides information on renal blood flow [3]; however, it only provides indirect macrovasculature parameters. Additionally, evaluation of cortical perfusion by US is challenging, particularly when the cortical blood flow is reduced. In these situations, US contrast agents can improve the diagnostic capabilities of conventional US, and allow the development of semi-quantitative and functional assessment of renal microvascular perfusion [3]. US contrast agents are superior to those used in CT or MRI for imaging of the vasculature because they behave the same as red blood cells and do not diffuse out of the vascular space [5]. Furthermore, these agents carry no risk of nephrotoxicity due to the absence of filtration and secretion by the kidneys. Contrast-enhanced ultrasound (CEUS) has been used as an excellent technique to assess renal parenchymal perfusion in patients with chronic kidney disease (CKD) [6, 7]. However, few studies have specifically assessed the utility of noninvasive evaluation of AKI using the CEUS technique.

The purpose of this study was to investigate quantitative CEUS parameters as hemodynamic predictors for renal outcomes in patients with AKI, in terms of the severity of AKI, initiation of renal replacement therapy (RRT), AKI recovery, and CKD progression.

## Materials and methods

### Study design and patients

This study was a prospective cohort study conducted between November 2017 and February 2019. Patients who were admitted or referred to the nephrology department in Incheon Saint Mary's Hospital because of a clinical diagnosis of AKI with varying degrees of renal dysfunction, and with different etiologies, were enrolled. AKI was diagnosed and staged using the serum creatinine concentration according to Kidney Disease Improving Global Outcomes (KDIGO) guidelines [8], and only those patients with known baseline serum creatinine levels within 6 months of enrollment were analyzed. Patients who were under 18 years of age, and those with contraindications to US contrast agents, such as a history of cardiac shunt, respiratory disorder, or hypersensitivity, were excluded. The evaluation proposal and draft data

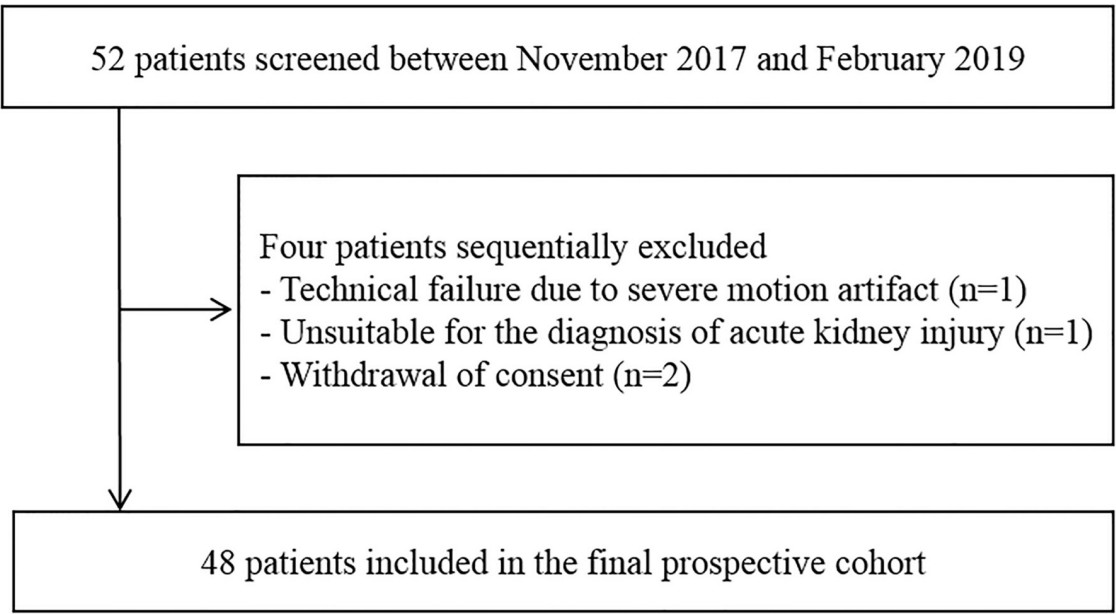

**Fig 1. Flow diagram of our study population.**

collection tools were reviewed and approved by the institutional review board of the Catholic University of Korea Catholic Medical Center on Oct 24, 2017. Informed written consent was obtained from all study participants (XC17BEDI0045). Fig 1 shows the flow diagram of the study population.

At the time of diagnosis of AKI, all subjects had blood samples drawn, serum creatinine and electrolyte concentrations assessed, and random urine samples collected for measurement of urine sodium, creatinine, and protein. Fractional excretion of sodium (FENa) was calculated using the following equation: [(urine sodium × serum creatinine)/(serum sodium × urine creatinine)] × 100. Baseline renal function was determined by the estimated glomerular filtration rate (eGFR), which was calculated using the Chronic Kidney Disease Epidemiology Collaboration equation [9]. Next, the CEUS examination was performed. All CEUS examinations and quantification analyses were performed by one experienced radiologist to ensure consistency among all measurements taken. The clinical context, history taking, physical examination, and interpretation of blood and urine laboratory findings were used for the differential diagnosis of the cause of AKI. Each patient was followed up for at least three months with regular blood tests, including serum creatinine. The follow-up interval was determined according to the physician's decision, and the frequency of follow-up differed between patients. Data consisting of patient demographics and comorbid conditions were also collected. Patients with underlying CKD were defined by evidence of impaired eGFR that had been present for > 3 months [10].

The primary outcome was the initiation of RRT. The secondary outcomes were AKI recovery and CKD progression. AKI recovery was defined as the return of serum creatinine to 25% of the baseline value. CKD progression was determined at three months after AKI and defined as new-onset proteinuria or a decline of 25% or more of eGFR compared to baseline eGFR. Baseline eGFR was defined as the most recent value before the diagnosis of AKI.

## US examination

All patients underwent CEUS with an iU 22 (Philips, Bothell, WA, USA), using a 1–5-MHz convex transducer. The contrast-specific imaging mode used in this study was pulse inversion harmonic imaging. The mechanical index was set at 0.06. We selected a maximum longitudinal scanning section that included the entire kidney. After identification of adequate images of the kidney, the transducer was manually held in the same scanning plane while patients were instructed to perform only shallow breathing to minimize the variation caused by motion. Then, an intravenous infusion of 2.0 mL of SonoVue (Bracco, Milan, Italy) was administrated via an antecubital vein in a bolus injection, followed by an immediate flush of 10 mL saline solution. The contrast agent was injected before examination of each kidney. The right kidney was examined first and, after about 20 min, the same CEUS examinations (including injection of SonoVue) were performed for the left kidney. Image depth, focus, gain, and frame rate were optimized and held constant for all further measurements. Continuous imaging was captured and observed in real-time for 5 min after SonoVue was injected. All images and video clips were stored digitally on a hard disk system, and then transferred to a personal computer for further quantitative analyses using advanced US quantification software (QLAB 8.1; Philips Medical Systems). The examination protocol is illustrated in Fig 2.

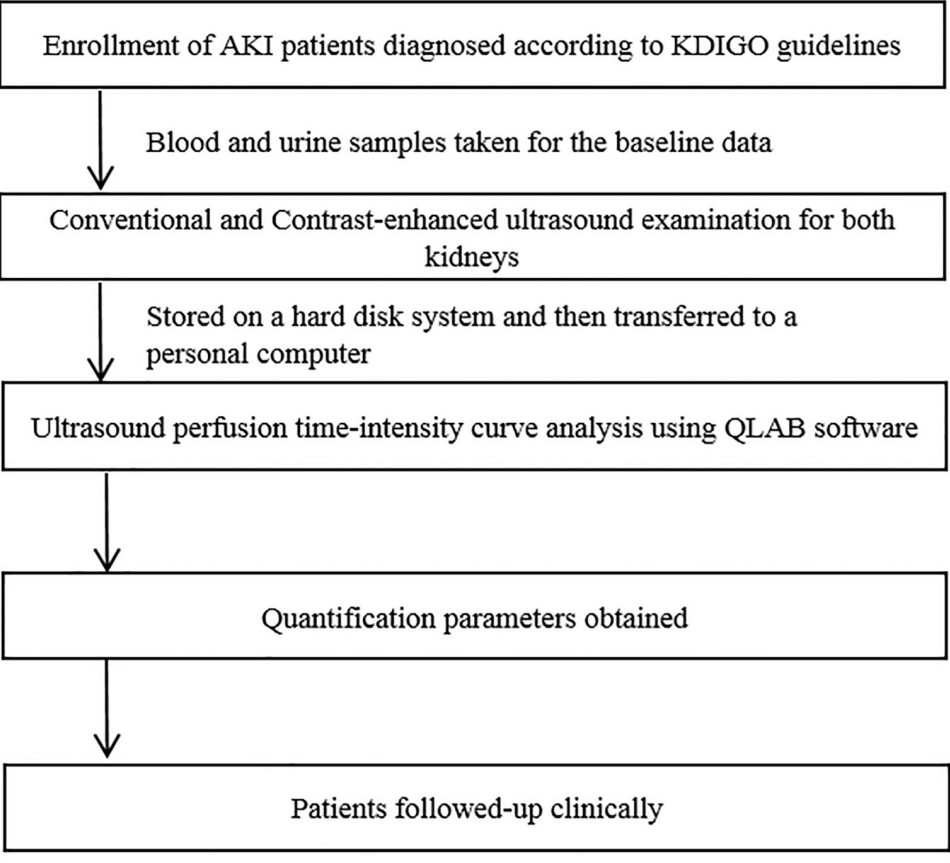

**Fig 2. Illustration of the examination protocol.**

## Image analysis

To compensate for minor breathing artifacts, all sequences were applied with motion compensation before the start of the analysis. Three similar-sized regions of interest (ROIs) ($5 \times 5$ mm$^2$) were drawn at the renal cortex and medulla, which are at the same approximate location and a similar depth, while excluding interlobar and arcuate arteries. In the ROI of each renal cortex and medulla, the computer-assisted program calculated acquisition of time (s) to signal intensity (dB) curves. The results of three ROIs at the renal cortex and medulla were averaged to minimize heterogeneity of the measurements. The data were then fitted to local density random walk wash-in and wash-out curves using the raw data. Fig 3 shows an example of the time–intensity curve (TIC) of the cortex (Fig 3A) and the medulla (Fig 3B).

The wash-in slope (WIS, the maximum wash-in velocity of the contrast medium; unit, dB/s), time to peak intensity (TTP, time to maximum enhancement; unit, s), peak intensity (PI, the maximum intensity of the curve; unit, dB), area under the TIC (AUC, the area under the TIC that was proportionate to the total volume of blood flow in the ROI; unit, dB), mean transit time (MTT, corresponding to the center of gravity of the perfusion model; unit, s), time for full width at half maximum (FWHM, time between the half amplitude values in each side of the maximum; unit, s), and rise time (RT, the time from injection until the peak of enhancement; unit, s) were obtained using QLAB software. For every ROI, the analysis was repeated three times, and the mean value of the perfusion parameter was obtained to minimize the transitional distance caused by respiration and to ensure the accuracy of the analyses. The final reported results of CEUS parameters represent the average value of each parameter from both kidneys of each individual.

## Statistical analysis

Values are expressed as means ± standard deviation and proportion of percent, as appropriate. Continuous data were compared using the Student's t-test or the Mann-Whitney U test, as appropriate. Pearson's correlation analysis or Spearman correlation analyses were used to determine the correlation between CEUS parameters and FENa or lowest urine output. We conducted a univariate logistic regression analysis to assess the impact of different US parameters on the clinical outcomes, including KDIGO AKI stage 3, need of RRT, AKI recovery, and CKD progression. Since CEUS parameters are not completely independent of one another, we included one CEUS parameter for each outcome in the logistic regression analysis. To further examine the predictive performance of CEUS parameters for clinical outcomes, receiver operating characteristic (ROC) curves were constructed to determine the optimal cutoff value. The intra-class correlation coefficient (ICC) was calculated to evaluate the intra-observer agreement [11]. The ICC was interpreted as follows: < 0.6 = poor; 0.6–0.79 = moderate; > 0.8–1 = excellent agreement. The differences in the CEUS parameters between patients with underlying CKD and those without CKD were also compared using independent Student's t-tests or Mann–Whitney U tests. All statistical analyses were conducted using SAS software (version 9.3, SAS Institute, USA). A $P$-value < 0.05 was considered statistically significant.

## Results

### Patient characteristics and CEUS-driven TIC parameters

A total of 48 consecutive patients with AKI (males, 25; females, 23; mean age, 60.65 ± 16.14 years) were enrolled. The baseline characteristics of patients are summarized in Table 1. Among them, 25 (52%) patients were diagnosed as KDIGO stage 3 AKI, 11 (23%) patients received RRT, 13 (27%) patients achieved functional recovery of AKI, and 18 (38%) patients

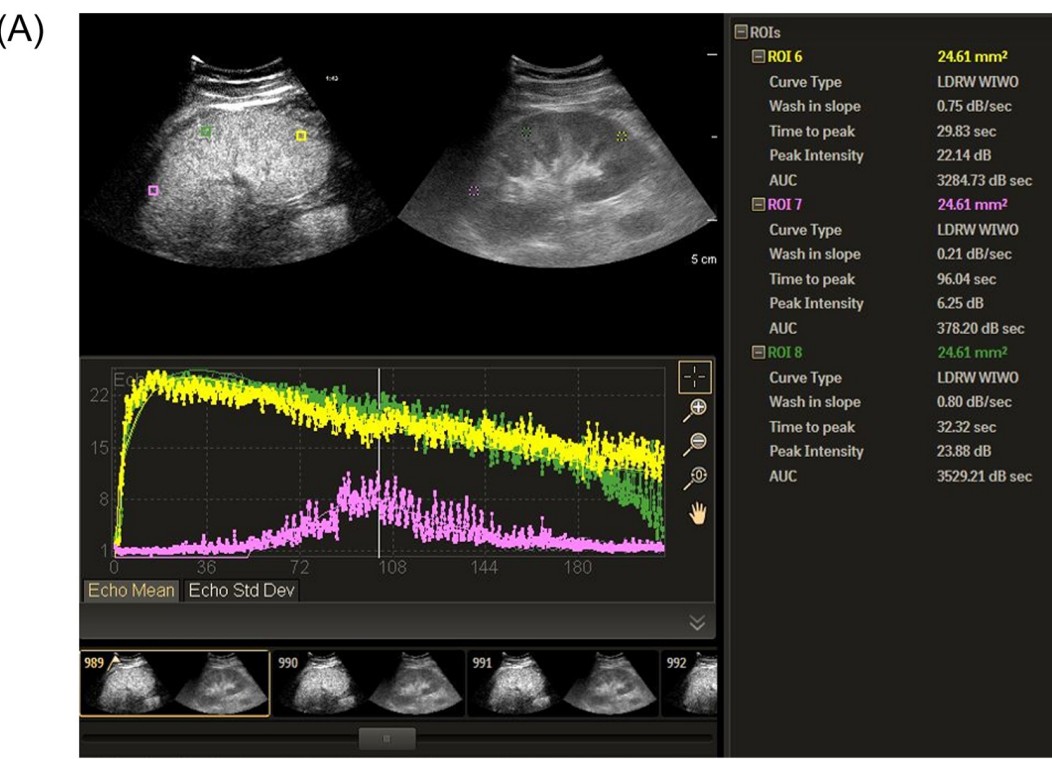

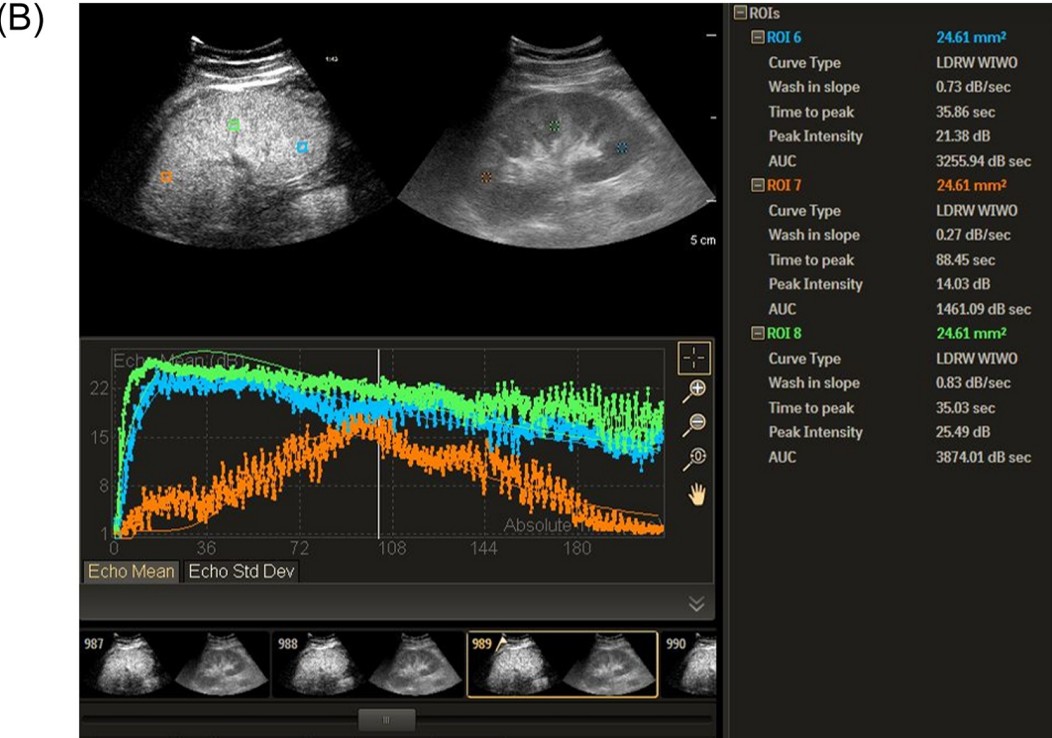

**Fig 3. Screenshot taken from QLAB illustrating localization of ROI and determining perfusion indices.** (A) Three ROIs were drawn in the renal cortex. The left part of the image shows contrast-image mode imaging and the right shows the standard (B-mode) imaging. (B) Three ROIs were drawn in the renal medulla. The left part of the image shows contrast-image mode imaging and the right shows the standard (B-mode) imaging. (A and B) Bottom: TIC curves. The smooth curves are the fitting curves, and the non-smooth curves are the original curves.

**Table 1. Baseline characteristics of the final trial cohort.**

| Characteristics | Value (n = 48) |
|---|---|
| Age (y)* | 61± 16 (25–85) |
| Sex | |
| Male | 25 (52%) |
| Female | 23 (48%) |
| Underlying renal disease | |
| No | 21 (44%) |
| Diabetes mellitus | 12 (26%) |
| Hypertension | 5 (11%) |
| Glomerulonephritis | 2 (4%) |
| Others | 8 (17%) |
| KDIGO AKI stage | |
| 1 | 11 (23%) |
| 2 | 12 (25%) |
| 3 | 25 (52%) |
| Cause of AKI | |
| Prerenal | 12 (25%) |
| Intrinsic | 34 (71%) |
| Postrenal | 1 (2%) |
| Intrinsic and postrenal | 1 (2%) |
| FeNa (%) * | 2.93 ± 4.27 (0.03–22.49) |
| Urine protein to creatinine ratio (mg/g) * | 3991 ± 7306 (164.3–45928) |
| Baseline eGFR (ml/min/1.73m$^2$) * | 64 ± 28 (13–127) |
| Serum creatinine at AKI occurrence (mg/dL)* | 4.01 ± 2.54 (1.30–14.70) |
| Highest serum creatinine (mg/dL)* | 4.46 ± 2.88 (1.60–14.70) |
| Lowest urine output (mL/day)* | 1221.20 ± 877.86 (0–3300) |

Unless otherwise indicated, data are number of patients.

*Values are mean ± SD with range in parentheses.

AKI = acute kidney injury, KDIGO = Kidney Disease: Improving Global Outcomes, FeNa = fractional excretion of sodium, eGFR = estimated glomerular filtration rate

showed CKD progression. Ten patients (20.8%) had urine output less than 500 mL per day, two patients (4.2%) developed anuria during the course of AKI, and the in-hospital mortality rate was 0%. Most of the causes of AKI were intrinsic (71%, n = 34), and prerenal causes ranked for the second common cause (25%, n = 12) (Table 1). Specific causes were as follows; drug, 42% (n = 20); infection, 17% (n = 8); glomerular disease, 13% (n = 6); gastrointestinal loss, 10% (n = 5); nephrolithiasis and benign prostate hyperplasia, 4% (n = 2); alcohol, 4% (n = 2); contrast, 2% (n = 1); hypotension, 2.1% (n = 1); others, 6% (n = 3).

No side-effects of the sonographic contrast agent were noted, and there was no hematuria or local pain during US examination. The duration between the time of CEUS and the time of peak serum creatinine was 3.8 ± 3.9 days (range, 0–14 days). The TIC parameters measured at the renal cortex and medulla are listed in Table 2.

The correlation coefficients between TIC parameters and FENa or lowest urine output were not statistically significant (S1 Table). The TIC parameters were compared between patients with intrinsic causes and those with prerenal or postrenal causes. None of the TIC parameters showed statistically significant difference between patients with intrinsic AKI and patients with prerenal or postrenal AKI (S2 Table).

**Table 2. The CEUS-driven TIS parameters of the final trial cohort.**

| TIC parameters | Value |
|---|---|
| Cortex | |
| WIS (dB/sec) | 0.93 ±1.00 (0.41–7.56) |
| TTP (s) | 43.13 ±11.14 (18.54–73.56) |
| PI (dB) | 17.66 ± 3.10 (10.12–24.49) |
| AUC (dB) | 2159 ± 497.4 (923.80–3061) |
| MTT (s) | 71.55 ± 15.84 (26.06–94.46) |
| FWHM (s) | 115.00 ± 25.37 (46.73–150.6) |
| RT (s) | 16.94 ± 4.50 (3.46–27.58) |
| Medulla | |
| WIS (dB/sec) | 0.89 ± 0.74 (0.42–4.52) |
| TTP (s) | 45.08 ± 11.95 (21.34–73.75) |
| PI (dB) | 17.80 ± 3.45 (8.97–25.37) |
| AUC (dB) | 2262 ± 542.30 (805.70–3192) |
| MTT (s) | 75.23 ± 13.54 (33.06–95.00) |
| FWHM (s) | 119.60 ± 25.53 (50.88–151.40) |
| RT (s) | 18.09 ± 5.00 (5.81–36.22) |

Values are mean ± SD with range in parentheses. TIC = time-intensity curve, CKD = chronic kidney disease, WIS = wash in slope, TTP = time to peak intensity, PI = peak intensity, AUC = area under the time-intensity curve, MTT = mean transit time, FWHM = time for full width half max, RT = rise time.

## Predictors of the severity of AKI and clinical outcomes and prognostic performance of CEUS

The univariate logistic regression analysis for the severity of AKI and clinical outcomes demonstrated that RT at the renal cortex (Odds ratio [OR], 1.21) predicted the KDIGO stage 3 AKI at the occurrence of AKI (Table 3). MTT and RT at the renal cortex (OR, 1.07 and 1.2, respectively) predicted the initiation of RRT. WIS and RT at the renal cortex and PI at the renal medulla (OR, 76.23, 0.83, and 1.25, respectively) predicted AKI recovery. In addition, PI and AUC at the renal medulla (OR, 0.78 and 1, respectively) predicted progression of CKD.

The ability of CEUS to predict KDIGO stage 3 AKI, initiation of RRT, AKI recovery, or CKD progression is shown in Table 4 by the AUC, sensitivity, specificity, and cut-off values. RT at the renal cortex showed reasonable prognostic performance for predicting KDIGO AKI stage 3 (AUC, 0.66, $P = 0.04$). MTT at the renal cortex was useful in predicting the initiation of RRT (AUC, 0.75, $P = 0.006$). WIS at the renal cortex and PI at the renal medulla were useful in predicting AKI recovery (AUC, 0.72 and 0.69, $P = 0.01$ and 0.04, respectively). PI and AUC at the renal medulla were also useful in predicting CKD progression (AUC, 0.73 and 0.7, $P = 0.003$ and 0.01, respectively); however, their sensitivity and specificity were low.

Fig 4 shows actual values of selected CEUS parameters for different AKI stages (Fig 4A), initiation of RRT (Fig 4B and 4C), AKI recovery (Fig 4D and 4E) and CKD progression (Fig 4F and 4G).

## Reproducibility of perfusion parameters

ICCs for quantitative TIC parameters were in the range of 0.26–0.98. The ICC for RT indicated poor agreement. The other parameters showed moderate-to-excellent agreement (Table 5).

**Table 3. Univariate logistic regression analysis of CEUS parameters.**

| TIC parameters | KDIGO AKI stage 3 | | Initiation of RRT | | AKI recovery | | CKD progression | |
|---|---|---|---|---|---|---|---|---|
| | OR (95% CI) | *P*-value | OR (95% CI) | *P*-value | OR (95% CI) | *P*-value | OR (95% CI) | *P*-value |
| Cortex | | | | | | | | |
| WIS (dB/sec) | 0.21 (0.01, 3.63) | 0.28 | 0.34 (0.01, 9.53) | 0.53 | 76.23 (1.47, 3955) | 0.03 | 0.14 (0.01, 2.94) | 0.21 |
| TTP (s) | 1.04 (0.98, 1.09) | 0.2 | 1.01 (0.95, 1.07) | 0.83 | 0.94 (0.89, 1.01) | 0.08 | 1.05 (0.99, 1.11) | 0.12 |
| PI (dB) | 0.98 (0.82, 1.18) | 0.87 | 0.98 (0.79, 1.23) | 0.89 | 1.21 (0.96, 1.51) | 0.1 | 0.84 (0.69, 1.04) | 0.11 |
| AUC (dB) | 1.00 (1.00, 1.00) | 0.46 | 1.00 (1.00, 1.00) | 0.5 | 1.00 (1.00, 1.00) | 0.3 | 1.00 (1.00, 1.00) | 0.34 |
| MTT (s) | 1.04 (1.00, 1.08) | 0.07 | 1.07 (1.01, 1.14) | 0.03 | 0.97 (0.92, 1.02) | 0.19 | 1.03 (0.99, 1.07) | 0.21 |
| FWHM (s) | 1.02 (0.99, 1.04) | 0.14 | 1.03 (0.99, 1.06) | 0.1 | 0.99 (0.96, 1.02) | 0.45 | 1.01 (0.98, 1.03) | 0.55 |
| RT (s) | 1.21 (1.03, 1.43) | 0.02 | 1.20 (1.00, 1.42) | 0.045 | 0.83 (0.70, 0.99) | 0.04 | 1.17 (1.00, 1.36) | 0.05 |
| Medulla | | | | | | | | |
| WIS (dB/sec) | 0.55 (0.18, 1.68) | 0.29 | 0.66 (0.16, 2.78) | 0.57 | 0.85 (0.38, 1.89) | 0.68 | 1.08 (0.49, 2.37) | 0.85 |
| TTP (s) | 1.01 (0.97, 1.07) | 0.55 | 1.01 (0.95, 1.06) | 0.85 | 0.99 (0.94, 1.05) | 0.8 | 1.00 (0.96, 1.06) | 0.86 |
| PI (dB) | 1.10 (0.92, 1.30) | 0.29 | 1.14 (0.91, 1.42) | 0.25 | 1.25 (1.02, 1.54) | 0.04 | 0.78 (0.63, 0.96) | 0.02 |
| AUC (dB) | 1.00 (1.00, 1.00) | 0.28 | 1.00 (1.00, 1.00) | 0.17 | 1.00 (1.00, 1.00) | 0.05 | 1.00 (1.00, 1.00) | 0.03 |
| MTT (s) | 1.02 (0.97, 1.06) | 0.46 | 1.05 (0.99, 1.12) | 0.12 | 0.99 (0.94, 1.04) | 0.67 | 1.01 (0.96, 1.05) | 0.8 |
| FWHM (s) | 1.01 (0.99, 1.04) | 0.42 | 1.02 (0.98, 1.05) | 0.33 | 1.00 (0.98, 1.03) | 0.88 | 1.00 (0.97, 1.02) | 0.87 |
| RT (s) | 1.03 (0.92, 1.16) | 0.6 | 1.13 (0.98, 1.30) | 0.1 | 1.03 (0.90, 1.18) | 0.64 | 0.95 (0.83, 1.07) | 0.38 |

AKI = acute kidney injury, KDIGO = Kidney Disease: Improving Global Outcomes, RRT = renal replacement therapy, CKD = chronic kidney disease, TIC = time-intensity curve, OR = odds ratio, CI = confidence intervals, WIS = wash in slope, TTP = time to peak intensity, PI = peak intensity, AUC = area under the time-intensity curve, MTT = mean transit time, FWHM = time for full width half max, RT = rise time.

## Comparison of TIC parameters between patients with CKD and those without CKD

Comparisons of the TIC parameters according to the presence of underlying CKD are shown in S3 Table. In patients with underlying CKD, PI and AUC were significantly decreased than in those without CKD in both the cortex and medulla. Also, the MTT and FWHM were shortened in patients with CKD in both the cortex and medulla compared to those without CKD.

**Table 4. Significant TIC parameters for predicting KDIGO AKI stage 3, initiation of RRT, AKI recovery, and CKD progression.**

| TIC parameters | AUC (95% CI) | *P*-value | Cut-off | Sensitivity (%) (95% CI) | Specificity (%) (95% CI) |
|---|---|---|---|---|---|
| KDIGO AKI stage 3 | | | | | |
| Cortex RT (s) | 0.66 (0.51, 0.82) | 0.04 | 17.05 | 60 (39, 79) | 70 (47, 87) |
| Initiation of RRT | | | | | |
| Cortex MTT(s) | 0.75 (0.57, 0.93) | 0.006 | 79.91 | 64 (31, 89) | 76 (59, 88) |
| Cortex RT (s) | 0.67 (0.48, 0.86) | 0.08 | 23.96 | 27 (6, 61) | 97 (86, 99) |
| AKI recovery | | | | | |
| Cortex WIS (dB/sec) | 0.72 (0.55, 0.9) | 0.01 | 0.657 | 83 (66, 93) | 62 (32, 86) |
| Medulla PI (dB) | 0.69 (0.51, 0.87) | 0.04 | 18.19 | 60 (42, 76) | 77 (46, 95) |
| CKD progression | | | | | |
| Medulla PI (dB) | 0.73 (0.58, 0.88) | 0.003 | 17.95 | 28 (10, 53) | 33 (17, 53) |
| Medulla AUC (dB) | 0.7 (0.55, 0.85) | 0.01 | 2369.11 | 17 (4, 41) | 47 (28, 66) |

TIC = time-intensity curve, AUC = area under receiver operating characteristic curve, CI = confidence interval, KDIGO = Kidney Disease: Improving Global Outcomes, RRT = renal replacement therapy, AKI = acute kidney injury, CKD = chronic kidney disease, WIS = wash in slope, PI = peak intensity, AUC = area under the time-intensity curve, MTT = mean transit time, RT = rise time.

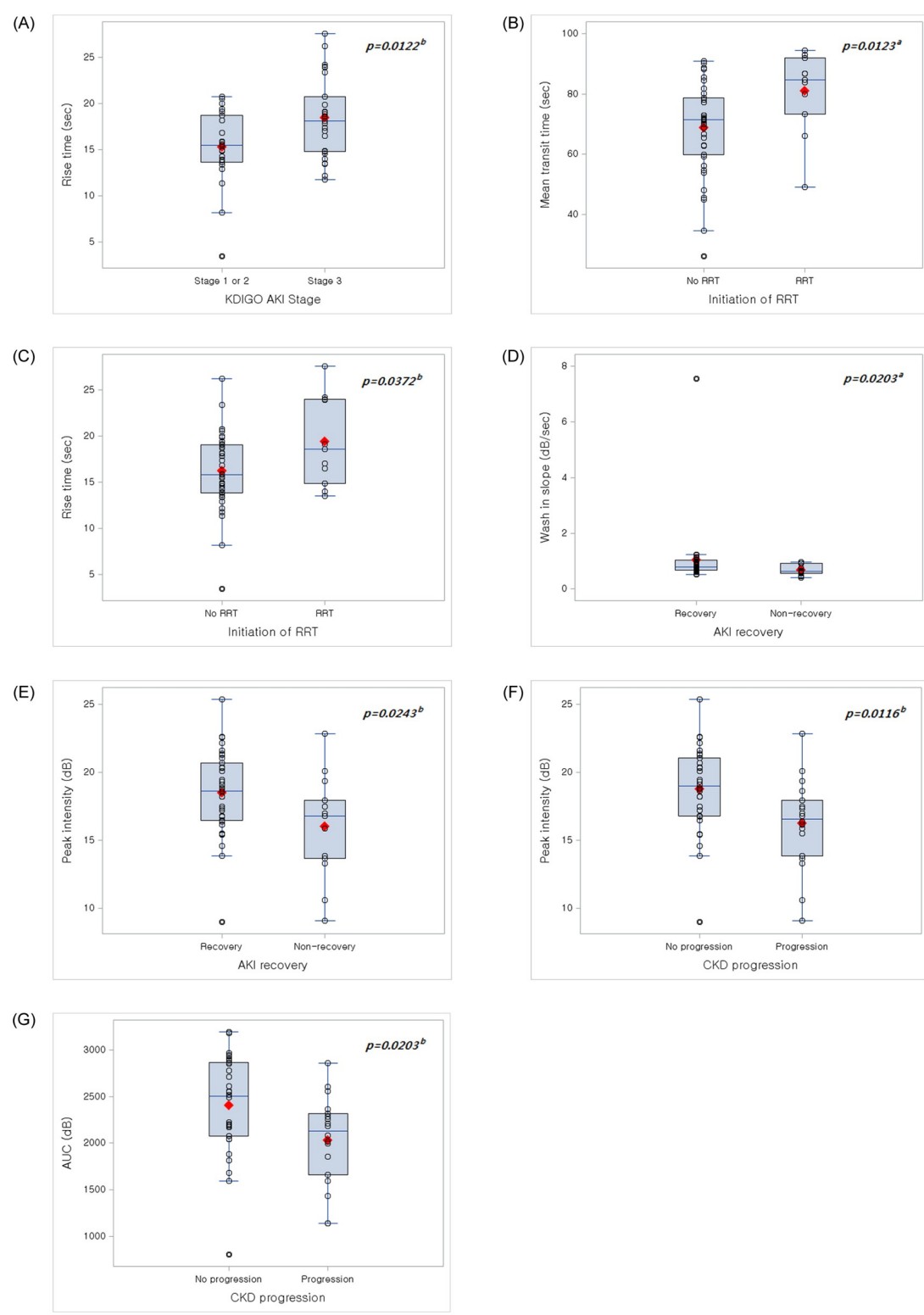

**Fig 4. Actual values of selected CEUS parameters associated with clinical outcomes.** (A) Cortex RT was significantly prolonged in KDIGO AKI stage 3 than in KDIGO AKI stage 1 or 2. (B and C) Cortex MTT and RT were significantly prolonged in patients who needed RRT than those who did not need RRT. (D and E) Cortex WIS and medulla PI were significantly higher in AKI recovery group than non-recovery group. (F and G) Medulla PI and AUC were significantly lower in patients with CKD progression than those without CKD progression. Vertical bars showed the median value of CEUS parameters in each group. The red diamonds showed the mean value. [a]$p<0.05$ by Mann-Whitney U test, [b]$p<0.05$ by Student's t-test.

**Table 5. Reproducibility of perfusion parameters.**

| TIC parameters | ICC (95% CI) | |
|---|---|---|
| | **Cortex** | **Medulla** |
| WIS (dB/sec) | 0.98 (0.96, 0.99) | 0.90 (0.85, 0.94) |
| TTP (s) | 0.60 (0.45, 0.73) | 0.68 (0.54, 0.79) |
| PI (dB) | 0.69 (0.56, 0.80) | 0.70 (0.57, 0.80) |
| AUC (dB/sec) | 0.64 (0.50, 0.76) | 0.76 (0.65, 0.84) |
| MTT (s) | 0.68 (0.55, 0.79) | 0.60 (0.45, 0.73) |
| FWHM (s) | 0.68 (0.55, 0.79) | 0.65 (0.51, 0.77) |
| RT (s) | 0.26 (0.12, 0.48) | 0.45 (0.29, 0.62) |

TIC = time-intensity curve, ICC = Intraclass correlation coefficient, CI = confidence interval, WIS = wash in slope, TTP = time to peak intensity, PI = peak intensity, AUC = area under the time-intensity curve, MTT = mean transit time, FWHM = time for full width half max, RT = rise time

## Discussion

In this prospective study, we identified several quantitative CEUS parameters that could be used as a predictor for renal outcomes in patients with AKI. These parameters included WIS, MTT, and RT at the renal cortex and PI and AUC at the renal medulla. Because the reproducibility of RT was poor, and the sensitivity and specificity of medullary AUC were low, based on their consistent reliability, we suggest MTT and WIS at the renal cortex and PI at the renal medulla for diagnosing the severity and predicting the renal prognosis in patients with AKI. Previous studies have shown that renal microcirculatory perfusion is impaired in animal models of AKI [12–14], and in humans with septic shock [15]. One animal study showed that perfusion impairment correlated with renal histological injury and CKD progression [14]. However, previous literature lacked any assessment of the association between CEUS-driven parameters and the clinical outcomes of human AKI, which is vital for the validity of CEUS for use in a clinical setting. To our knowledge, this study is the first to present the clinical application of CEUS to assess renal prognosis in patients with AKI.

Generally, the diagnosis of AKI is based on changes in serum creatinine concentration, but these changes poorly reflect the acute deterioration in renal function [16], and serum creatinine levels lack sensitivity and specificity, resulting in higher rates of delayed and missed diagnosis [17, 18]. Therefore, the search for new urinary and serum biomarkers, which have the potential to provide earlier diagnosis and better prognosis, is ongoing [19]. Imaging techniques usually provide information concerning the anatomy of the kidney, the possibility of obstruction, inflammation, and edema [3]. Traditionally, Doppler US has been considered as a potential imaging technique to detect renal blood perfusion abnormalities [20]. However, resistive index values only correlate with macroangiopathy and might be influenced by factors such as increased intra-abdominal pressure, pulse rate, pharmacotherapy, and the site at which it is measured [21]. Clinical use of Doppler US is limited by its lower detection limit, the inability to detect slow flow velocity, and limited accuracy in quantifying renal blood flow. CEUS is a promising tool that can be used as a noninvasive approach without the added risks of ionizing radiation and nephrotoxicity, which would impair renal perfusion and increase the risk of nephrogenic systemic fibrosis [22]. The changes in perfusion indices driven by CEUS parallel those in effective renal plasma flow [23]. In addition, in contrast to standard serum markers of renal function, it is possible to obtain a map of the kidney microvasculature with high temporal and spatial resolution [24]. Furthermore, CEUS is a relatively uncomplicated procedure that can be applied to critically ill patients [15, 25]. In a study using CEUS in patients with septic shock, the decreased cortical

perfusion, which manifested as lower PI and higher MTT, was associated with severe AKI [15]. Similarly, in our study, the cortical MTT predicted the initiation of RRT. Other cortical parameters also showed meaningful results; for example, cortical RT predicted KDIGO stage 3 AKI and initiation of RRT, and cortical WIS predicted AKI recovery. Given the poor reproducibility of RT, we speculate that cortical MTT and WIS can be used to predict AKI outcomes.

Most of the CEUS measurements that were used to monitor renal microcirculatory perfusion provided information on cortical perfusion [23, 25]. Most recently, it was reported that medullary hypoxia due to intrarenal blood flow redistribution is important in the development of AKI [26–28]. Therefore, we assessed medullary perfusion and found several predictive parameters: medullary PI predicted AKI recovery and CKD progression, and medullary AUC predicted CKD progression. Because the sensitivity and specificity of medullary PI and AUC were low for predicting CKD progression, we speculate that medullary PI may be a useful parameter to predict AKI recovery. In this study, the CEUS-driven TIC parameters of the medulla were not reduced compared to those of cortex. This was unexpected, as the renal medulla receives lower blood flow than the cortex [29]. We speculate that a medullary blood congestion and consequent slowing of the blood flow occurred in our patients with AKI. This medullary congestion is a hallmark of ischemic AKI in both experimental models [30–33] and in specimens obtained from biopsy or at autopsy [34]. Currently, there is no CEUS study for medullary perfusion in humans with AKI. A recent animal study showed that medullary PI and AUC were about one-third of that of cortex in healthy dogs, on the other hand, medullary PI and AUC were almost the same as cortex values in dogs with AKI [12]. The increase in medullary PI and AUC seen in dogs with AKI, which suggests increased medullary congestion, is similar to our results. Further studies with a larger sample size are needed for validation of medullary CEUS findings in patients with AKI.

Our study found that the MTT at the renal cortex was increased in patients requiring RRT than that of patients not requiring RRT. Considering that MTT indicates the average time taken by blood to pass through the capillary network, this finding indicated that, less contrast microbubbles entered the renal cortex microvascular bed with slow perfusion in unit time in patients requiring RRT compared with those not requiring RRT. Harrois *et al* showed that the greater the alteration in MTT is, the higher is the risk of severe AKI, indicating that MTT seems to be mostly linked to intrarenal hemodynamics [15]. PI reflects the quantity of contrast agent microbubbles in the vascular bed of the organ, while WIS reflects the early quantity and velocity during contrast agent perfusion. These two parameters are associated with the degree of vascularization. The WIS at the renal cortex and the PI at the renal medulla was higher in the AKI recovery group than that in the non-recovery group. This finding suggests that better vascularization at the renal cortex and medulla is associated with AKI recovery. In addition, lower PI at the renal medulla was associated with CKD progression, which meant that reduced medullary perfusion is not protective against tissue recovery. This finding is consistent with a previous report which showed that PI at the renal medulla decreased as the CKD stages progressed [35]. Increased RT indicates delay in rise in echogenicity, which is related with increased resistance of glomeruli and peritubular capillaries [35]. In this study, increased RT at the cortex was associated with severe AKI stage and need of RRT. This finding appears to reflect the delayed cortical perfusion of patients with severe AKI. The AUC means the total volume of blood flow. In our study, the increase in AUC at renal medulla, which indicates medullary blood congestion, was associated with CKD progression. It is difficult to analyze the clinical implication of this finding, since the AUC is dependent on the capillary resistance, retention time and total capillary volume. We can only speculate that an alteration in the medullary capillary density may be associated with progressive renal injury. However, since the sensitivity, specificity, and reliability of medullary PI, AUC, and RT were low, the clinical significance of these parameters needs caution in interpretation.

Quantitative analysis showed that cortical and medullary PI and AUC, intensity- and blood volume-related parameters, were decreased and that cortical and medullary MTT and FWHM, timed-related parameters, were shortened, according to the presence of CKD. Therefore, in CKD patients, renal contrast enhancement was attenuated with deterioration of the renal function, which is consistent with a previous study [35]. Additionally, these results suggest that the CEUS parameters may be used to discriminate between those patients with AKI and those with AKI on CKD.

It should be noted that CEUS should not replace serum markers of renal function, but it might prove useful as a diagnostic tool in patients with inconclusive clinical, laboratory, and histological findings. The early identification of patients who will need RRT and who are at a high risk of progression to CKD would assist physicians in planning and initiating the appropriate management to improve renal outcomes, and to develop renal-preserving treatments. Furthermore, the ability to visualize and quantify changes in microperfusion could provide useful supplemental information for additional investigations of disease mechanisms or novel therapeutic strategies.

Our study had some limitations. First, it was a single-center study with a relatively small number of patients. It was difficult to enroll patients in severe illness, which explains the small number of enrollment and the reason for the low number of patients who developed anuria and the low in-hospital mortality rate. As the numbers of patients and outcome events were small, we could not perform a multivariate logistic analysis. Therefore the predictability of the CEUS perfusion parameters cannot be generalized yet. Second, the time of CEUS examination varied between patients, and the CEUS was performed during different stages of AKI evolution. However, it was designed as a pilot trial to investigate the feasibility of quantitative CUES analysis for predicting renal outcomes in AKI patients. Further studies with a larger sample size and follow-up investigations need to be carried out for further validation. Third, the causes of AKI were various, and analyses according to the cause of AKI were not carried out due to the small sample size. Different causes of AKI may alter the hemodynamics and affect the final quantitative results. The high variability of urine protein-to-creatinine ratio results from this various etiology of AKI. Fourth, heterogeneity of measurements is a common limitation in all attempts to use CEUS to quantify organ perfusion. To minimize this parameter, we aimed to locate ROIs at similar depth and distance, as recommended by Averkiou et al. [36]. In addition, three ROIs were drawn for each experimental time point, and the results were averaged to minimize heterogeneity of the measurements. Fifth, we cannot completely exclude the possibility that concomitant medication usage confounded our results. Kidney perfusion can be influenced by medications and hydration status [37].

In conclusion, we observed several alterations in the CEUS perfusion parameters that showed significance in predicting the severity of AKI and renal prognosis. By evaluating renal microvascular perfusion, CEUS may be used as a supplemental tool to estimate the severity of renal dysfunction and to predict renal outcomes after AKI.

## Supporting information

**S1 Table. Correlations between TIC parameters and FENa or the lowest urine output.**
(DOCX)

**S2 Table. Comparison of TIC parameters between patients with intrinsic AKI and patients with prerenal or postrenal AKI.**
(DOCX)

**S3 Table. Difference of TIC parameters according to the presence of underlying CKD.**
(DOCX)

## Acknowledgments

The Statistical consultation was supported by the Department of Biostatistics of the Catholic Research Coordinating Center.

## Author Contributions

**Conceptualization:** Hye Eun Yoon, Yu Ri Shin.

**Data curation:** Hye Eun Yoon, Da Won Kim, Dongryul Kim, Yaeni Kim, Seok Joon Shin, Yu Ri Shin.

**Formal analysis:** Hye Eun Yoon, Yu Ri Shin.

**Funding acquisition:** Yu Ri Shin.

**Software:** Yu Ri Shin.

**Writing – original draft:** Hye Eun Yoon, Yu Ri Shin.

**Writing – review & editing:** Hye Eun Yoon, Yu Ri Shin.

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
