## [Decision Letter · Decision Letter 0]

8 Apr 2020

PONE-D-20-06286

A pilot trial to evaluate the clinical usefulness of contrast-enhanced ultrasound in predicting renal outcomes in patients with acute kidney injury

PLOS ONE

Dear Dr. Shin,

Thank you for submitting your manuscript to PLOS ONE. After careful consideration, we feel that it has merit but does not fully meet PLOS ONE’s publication criteria as it currently stands. Therefore, we invite you to submit a revised version of the manuscript that addresses the points raised during the review process.

Two out of three experts raised several concerns on method and analysis.

We would appreciate receiving your revised manuscript by May 23 2020 11:59PM. To enhance the reproducibility of your results, we recommend that if applicable you deposit your laboratory protocols in protocols.io, where a protocol can be assigned its own identifier (DOI) such that it can be cited independently in the future. For instructions see: http://journals.plos.org/plosone/s/submission-guidelines#loc-laboratory-protocols

We look forward to receiving your revised manuscript.

Kind regards,

Tatsuo Shimosawa, M.D., Ph.D.

Academic Editor

PLOS ONE

Reviewers' comments:

Reviewer's Responses to Questions

**Comments to the Author**

1. Is the manuscript technically sound, and do the data support the conclusions?

Reviewer #1: Partly

Reviewer #2: Yes

Reviewer #3: Yes

2. Has the statistical analysis been performed appropriately and rigorously? 

Reviewer #1: No

Reviewer #2: Yes

Reviewer #3: Yes

3. Have the authors made all data underlying the findings in their manuscript fully available?

Reviewer #1: Yes

Reviewer #2: Yes

Reviewer #3: Yes

4. Is the manuscript presented in an intelligible fashion and written in standard English?

Reviewer #1: Yes

Reviewer #2: Yes

Reviewer #3: Yes

5. Review Comments to the Author

Reviewer #1: Dear Editor,

I reviewed the manuscript entitled, ‘A pilot trial to evaluate the clinical usefulness of contrast-enhanced ultrasound in predicting renal outcomes in patients with acute kidney injury’ by Shin and collegaues.

I have the following comments and questions,

1. Since CEUS is not a widely used technology and even when it is used, the methodology varies widely, I recommend a brief description of different parameters and what each of them indicate before proceeding to further discussion of methodology. For example, what does WIS or MTT after an injection of a bolus of contrast media indicate and what do we expect to see in various forms of AKI.

2. Estimated GFR by CKD Epi formula should only be used to determine the baseline renal function and not after AKI.

3. It would be helpful to know the cause of AKI and the urine output in these patients. Data on the number of patients requiring RRT and a scale of severity of illness would also be valuable. I am surprised that mortality was zero.

4. Do CEUS variables differentiate AKI causes or correlate with FENa or FEUrea or response to volume expansion?

5. I am not sure if logistic regression is the right test, considering CEUS parameters are not completely independent of one another. If you were including one CEUS variable in the model and were adding other potential predictors of AKI outcome, that might have been useful.

6. Considering medulla receives only about 15% of renal blood flow and has very slow velocities and transition times, I am very surprised to see comparable numbers for cortex and medulla. The time to peak is almost identical for the cortex and medulla.

7. In the image provided the ROI are much smaller than what we use and other investigators have used. I am wondering why larger ROI are not used, at least for the cortex.

8. The image intensity needs to be standardized for the renal artery, aorta or left ventricle intensity. That would eliminate the issues with errors in volume of contrast, extravasation from the vein, etc.

9. None of the parameters from CEUS have good AUC or reasonable sensitivity or specificities.

10. Under discussion, I recommend more details on interpretation of results. In other words, it would be helpful to know why higher or lower values of certain parameters indicate a more severe degree of tissue injury. What is the clinical implication of these findings?

Reviewer #2: Dear Prof. Tatsuo Shimosawa,

Academic Editor,

PLOS One

The manuscript of “A pilot trial to evaluate the clinical usefulness of contrast-enhanced ultrasound in predicting renal outcomes in patients with acute kidney injury” is interesting to develop and establish renal vessel CEUS. I expect the new method is important for evaluation of real-time renal function of AKI as well as CKD.

I have several questions about methods.

Comments;

1) There are a lot of unique abbreviations. I recommend to make abbreviation section.

2) I recommend to give a figure of the examination protocol. It is useful to understand the examination and the result data.

3) How many sonographers did exam the protocol?

4) It was written “the transducer was manually held in the same scanning plane while patients were instructed to perform only shallow breathing to minimize the variation caused by motion” in Materials and Methods. I think each holding time was very long because the protocol time should include baseline, bolus shot and follow up. And I concern the base line data should get much different when the angle of the transducer moves just a little. How they contrive ways to keep the transducer in the same place and the same angle?

5) There is not section of COI nor funding.

Reviewer #3: Summary of the research: This is a prospective study on CEUS in 48 patients with AKI. CEUS parameters measured included: wash-in slope (WIS), time to peak intensity, peak intensity (PI), area under the time-intensity curve (AUC), mean transit time (MTT), time for full width at half maximum, and rise time (RT). The outcomes assessed included KDIGO AKI stage, initiation of RRT, AKI recovery, and CKD. Significant findings included: Cortical rise time predicted KDIGO stage 3 AKI, Cortical WIS and RT predicted need for RRT. Cortical WIS and medullary PI predicted AKI recovery. Medullary PI and AUC predicted CKD progression. Because of the small patient population, subtypes of AKI were not able to be evaluated. Intraobserver evaluation with multiple ROIs was performed.

Overall Impression: The strengths of this article are many. Novelty, prospective study design, relatively equal split between patient genders, and the incorporation of intraobserver evaluation for quality control. I think that the authors did a great job of setting the groundwork for future studies with this small pilot study and were very open and realistic with study limitations (many of their limitations are understandable given the patient population and commonly seen with other renal studies such as inability to account for medications impacting renal function at time of study).

Major Issues: I do not see any major issues with the study as performed or the paper as written. I think the authors have done a wonderful and thoughtful job.

Minor Issues: None.

Miscellaneous Remarks: Valuable addition to the literature.

6. PLOS authors have the option to publish the peer review history of their article (what does this mean?). If published, this will include your full peer review and any attached files.

Reviewer #1: No

Reviewer #2: No

Reviewer #3: Yes: Linda C Kelahan

---

## [Author Response · Author response to Decision Letter 0]

24 Apr 2020

To the editor

We would like to thank the editor and reviewers for their insightful comments. Their comments definitely helped to improve the quality of our manuscript. We have tried our very best to answer the reviewers’ specific points. The reviewer's comments are in blue and our answers are in black. We hope our revisions improve the paper such that it is worthy of publication in PLOS ONE. Thank you. 

Point by point response to reviewers’ comments

[1] Journal requirements:

When submitting your revision, we need you to address these additional requirements. Please ensure that your manuscript meets PLOS ONE's style requirements, including those for file naming.

In response to this comment, we made sure that our manuscript was correctly formatted, according to the PLOS ONE's style requirements.

[2] Reviewer #1 comments: 

1. Since CEUS is not a widely used technology and even when it is used, the methodology varies widely, I recommend a brief description of different parameters and what each of them indicate before proceeding to further discussion of methodology. For example, what does WIS or MTT after an injection of a bolus of contrast media indicate and what do we expect to see in various forms of AKI.

We thank the reviewer for this helpful comment. However, there is limit to the number of characters in the text and how the CEUS parameters appear in various forms of AKI is unknown yet. We can only speculate what these parameters mean in clinical settings. To avoid redundancy, we additionally described the meanings and clinical implications of the quantitative CEUS parameters in the discussion section.

2. Estimated GFR by CKD Epi formula should only be used to determine the baseline renal function and not after AKI.

Thank you for the comment. We deleted the “eGFR at AKI occurrence” and the “lowest eGFR” from Table 1. Instead we included the “serum creatinine at AKI occurrence” and the “highest serum creatinine” in Table 1. 

3. It would be helpful to know the cause of AKI and the urine output in these patients. Data on the number of patients requiring RRT and a scale of severity of illness would also be valuable. I am surprised that mortality was zero.

We added the causes of AKI (prerenal, intrinsic, and postrenal) in Table 1 and also described it in more detail in the result section. The lowest urine output was added in Table 1. The AKI severity was shown as the KDIGO AKI stage in Table 1 and was described in the result section as “Among them, 25 (52%) patients were diagnosed as KDIGO stage 3 AKI, 11 (23%) patients received RRT, 13 (27%) patients achieved functional recovery of AKI, and 18 (38%) patients showed CKD progression. Ten patients (20.8%) had urine output less than 500 mL per day, two patients (4.2%) developed anuria during the course of AKI, and the in-hospital mortality rate was 0%.”

As the reviewer commented, the mortality rate was zero. This was because it was difficult to enroll patients in severe illness, since patients who are hemodynamically unstable and are in a clinically downhill course cannot undergo CEUS. This was described in the limitation. 

4. Do CEUS variables differentiate AKI causes or correlate with FENa or FEUrea or response to volume expansion?

As the reviewer suggested, we analyzed the correlation coefficient between the CEUS variables and FENa or urine output. None of the CEUS variables showed statistically significant correlation coefficients (supplement table 1). The data on FEUrea or response to volume expansion was not collected in this study. 

The CEUS variables were compared between patients with intrinsic causes and those with prerenal or postrenal causes. None of the TIC parameters showed statistically significant difference between patients with intrinsic AKI and patients with prerenal or postrenal AKI (Supplement table 2). Therefore the CEUS variables did not differentiate AKI causes. 

The description for supplement tables 1 and 2 were added in the result section. 

5. I am not sure if logistic regression is the right test, considering CEUS parameters are not completely independent of one another. If you were including one CEUS variable in the model and were adding other potential predictors of AKI outcome, that might have been useful.

The statistical analysis in our study was performed by statistician in our university. We consulted with the statistician about performing the multivariate analysis, and received this answer as follows. 

“In statistics, the ‘one in ten rule’ is a rule of thumb for how many predictor parameters can be estimated from data when doing regression analysis, while keeping the risk of overfitting low. The rule states that one predictive variable can be studied for every ten events. In this study, the outcome events were need of RRT, AKI recovery, and CKD progression. The number of the events was small, which was all less than 20. Therefore, to keep the overfitting low and make the result reliable, only one predictor can be included in the logistic regression analysis. That is why we only performed a univariate logistic regression analysis”.

We agree with the reviewer’s comment that a multivariate analysis is needed to demonstrate the predictive value of CEUS parameters. But as the number of patients and number of outcome events were small, we could not perform a multivariate analysis. We added this as a limitation in the discussion section. 

“…As the numbers of patients and outcome events were small, we could not perform a multivariate logistic analysis. Therefore the predictability of the CEUS perfusion parameters cannot be generalized yet…..”

6. Considering medulla receives only about 15% of renal blood flow and has very slow velocities and transition times, I am very surprised to see comparable numbers for cortex and medulla. The time to peak is almost identical for the cortex and medulla.

Currently, there is no CEUS study for medullary perfusion in human. A recent animal study showed that medullary PI and AUC were about one-third of that of cortex in healthy dogs, on the other hand, medullary PI and AUC were almost the same as cortex values in dogs with AKI (J Small Anim Pract. 2019 Aug;60(8):471-476). The increase in medullary PI and AUC seen in dogs with AKI, which suggests increased medullary perfusion, is similar to our results. We speculate that a medullary blood congestion and consequent slowing of the blood flow occurred in our patients with AKI. This medullary congestion was shown to be a key event in ischemic AKI murine model (Kidney Int. 1984 Sep;26(3):283-93).

7. In the image provided the ROI are much smaller than what we use and other investigators have used. I am wondering why larger ROI are not used, at least for the cortex.

According to the referenced literature (Current Drug Targets 2009;10(12):1184-9), the pattern and quantity of blood flow and tissue blood volume in different regions within the organ, i.e., cortex separately from medulla, can be determined within a few minutes with CEUS, which is particularly desirable in clinical situations in which patient transfer out of the intensive care unit is an issue. By providing information on regional changes in renal blood flow, CEUS has the potential to serve as a helpful tool in the diagnostic work up of patients with AKI.

We wanted to find out if there is a perfusion difference between cortex and medulla or to look at corticomedullary perfusion ratios of CEUS parameters. Therefore, we thought that the ROI of the cortex and medulla should be drawn in the same size and shape as possible. However, the medulla, unlike a cortex, is not easy to draw an entire ROI. So, we analyzed by drawing three ROIs of the same size in the cortex and medulla in the upper, middle, and lower regions. In order to include only the microvasculature in the outer cortex according to very strict criteria, many reported data obtained the ROI of the cortex in the same way as in our study.

8. The image intensity needs to be standardized for the renal artery, aorta or left ventricle intensity. That would eliminate the issues with errors in volume of contrast, extravasation from the vein, etc.

We agree with to the reviewer's comment, in that a suitable arterial input function is usually not available in the bolus-administration mode. In addition, some perfusion parameters are severely influenced by ultrasound attenuation in intervening tissue. However, the heart, renal parenchyma and bowel wall represent the most established applications of CEUS in the quantification of organ perfusion in humans. Until recently, the quantitative analysis of renal tissue perfusion, measured by CEUS, has been performed without quantification standardization. In addition, although manual injection might potentially predispose to lower accuracy in contrast infusion, the majority of studies used manual injection. Therefore, we did not standardize the image intensity of the renal parenchyme to renal artery, aorta or left ventricle intensity.

9. None of the parameters from CEUS have good AUC or reasonable sensitivity or specificities.

In our study, we found that the AUC in MTT, WIS, PI values were 0.75, 0.72, and 0.69, respectively. Although diagnostic accuracy is not satisfactory, our preliminary findings are encouraging for the performance of quantitative CEUS analysis as a screening test for patients with AKI. Many factors can affect CEUS quantitative analysis, including the frequency, the mechanical index, the analytical software, the contrast dose, patient’s factor, the injection velocity of contrast agents and so on. In this regard, further standardization and investigation are needed.

10. Under discussion, I recommend more details on interpretation of results. In other words, it would be helpful to know why higher or lower values of certain parameters indicate a more severe degree of tissue injury. What is the clinical implication of these findings?

We thank the reviewer for this helpful comment. In response to this comment, we added the following sentence in discussion:

“Our study found that the MTT at the renal cortex was increased in patients requiring RRT than that of patients not requiring RRT. Considering that MTT indicates the average time taken by blood to pass through the capillary network, this finding indicated that, less contrast microbubbles entered the renal cortex microvascular bed with slow perfusion in unit time in patients requiring RRT compared with those not requiring RRT. Harrois et al showed that the greater the alteration in MTT is, the higher is the risk of severe AKI, indicating that MTT seems to be mostly linked to intrarenal hemodynamics (Crit care 2018;22:161). PI reflects the quantity of contrast agent microbubbles in the vascular bed of the organ, while WIS reflects the early quantity and velocity during contrast agent perfusion. These two parameters are associated with the degree of vascularization. The WIS at the renal cortex and the PI at the renal medulla was higher in the AKI recovery group than that in the non-recovery group. This finding suggests that better vascularization at the renal cortex and medulla is associated with AKI recovery. In addition, lower PI at the renal medulla was associated with CKD progression, which meant that reduced medullary perfusion is not protective against tissue recovery. This finding is consistent with a previous report which showed that PI at the renal medulla decreased as the CKD stages progressed (Int Heart J 2010;51:176-82). Increased RT indicates delay in rise in echogenicity, which is related with increased resistance of glomeruli and peritubular capillaries (Int Heart J 2010;51:176-82). In this study, increased RT at the cortex was associated with severe AKI stage and need of RRT. This finding appears to reflect the delayed cortical perfusion of patients with severe AKI. The AUC means the total volume of blood flow. In our study, the increase in AUC at renal medulla, which indicates medullary blood congestion, was associated with CKD progression. It is difficult to analyze the clinical implication of this finding, since the AUC is dependent on the capillary resistance, retention time and total capillary volume. We can only speculate that an alteration in the medullary capillary density may be associated with progressive renal injury. However, since the sensitivity, specificity, and reliability of medullary PI, AUC, and RT were low, the clinical significance of these parameters needs caution in interpretation.”

[3] Reviewer #2 comments: 

1. There are a lot of unique abbreviations. I recommend to make abbreviation section.

As the reviewer commented, we added an abbreviation section in the end of the manuscript. 

2. I recommend to give a figure of the examination protocol. It is useful to understand the examination and the result data.

In response to this comment, we made a figure of the examination protocol and added the sentence in the material and methods:

“The examination protocol is illustrated in Figure 2.”

3. How many sonographers did exam the protocol?

All CEUS examinations were performed by one experienced radiologist to ensure the stability of measurements.

This sentence has already been included in the Materials and Methods.

4. It was written “the transducer was manually held in the same scanning plane while patients were instructed to perform only shallow breathing to minimize the variation caused by motion” in Materials and Methods. I think each holding time was very long because the protocol time should include baseline, bolus shot and follow up. And I concern the base line data should get much different when the angle of the transducer moves just a little. How they contrive ways to keep the transducer in the same place and the same angle?

Examinations were performed as followed: First, conventional B-mode US was performed to scan the kidney and to observe the size, position and echogenicity. During B-mode scanning, maximal longitudinal scanning was chosen as the ideal plane for CEUS. Second, the US system was then switched to contrast mode for CEUS examination. Contrast agent was injected manually as a bolus over 2 seconds followed by a flush of 10 mL saline solution. The entire CEUS process was recorded for each patient from the time of injection, until no apparent agent was observed. Our study analyzed images of kidney perfusion after the contrast agent was given, focusing only on cortex and medulla perfusion. Conventional ultrasound examination is intended to confirm the approximate target organ location for contrast-enhanced ultrasound examination to be performed later, and there is no special baseline data. In addition, the kidney is an organ located in the retroperitoneal space, and unlike other organs in the abdominal cavity, there is less movement caused by patient's breathing. In addition, we specifically instructed the patients regarding their respiration control before the CEUS examination was performed. We found that our study showed a relatively stable perfusion graph except for patients with severe respiratory disease.

5. There is not section of COI nor funding.

We entered a financial disclosure statement and COI statement during the submission process.

[4] Reviewer #3 comments: 

Thank you for your informative comments.

---

## [Decision Letter · Decision Letter 1]

11 May 2020

PONE-D-20-06286R1

A pilot trial to evaluate the clinical usefulness of contrast-enhanced ultrasound in predicting renal outcomes in patients with acute kidney injury

PLOS ONE

Dear Dr. Shin,

Thank you for submitting your manuscript to PLOS ONE. After careful consideration, we feel that it has merit but does not fully meet PLOS ONE’s publication criteria as it currently stands. Therefore, we invite you to submit a revised version of the manuscript that addresses the points raised during the review process.

Two experts and I have concern on statistical analysis and data collection method.

We would appreciate receiving your revised manuscript by Jun 25 2020 11:59PM. To enhance the reproducibility of your results, we recommend that if applicable you deposit your laboratory protocols in protocols.io, where a protocol can be assigned its own identifier (DOI) such that it can be cited independently in the future. For instructions see: http://journals.plos.org/plosone/s/submission-guidelines#loc-laboratory-protocols

We look forward to receiving your revised manuscript.

Kind regards,

Tatsuo Shimosawa, M.D., Ph.D.

Academic Editor

PLOS ONE

Reviewers' comments:

Reviewer's Responses to Questions

**Comments to the Author**

1. If the authors have adequately addressed your comments raised in a previous round of review and you feel that this manuscript is now acceptable for publication, you may indicate that here to bypass the “Comments to the Author” section, enter your conflict of interest statement in the “Confidential to Editor” section, and submit your "Accept" recommendation.

Reviewer #1: All comments have been addressed

Reviewer #2: (No Response)

Reviewer #3: All comments have been addressed

2. Is the manuscript technically sound, and do the data support the conclusions?

Reviewer #1: Yes

Reviewer #2: Partly

Reviewer #3: Yes

3. Has the statistical analysis been performed appropriately and rigorously? 

Reviewer #1: No

Reviewer #2: Yes

Reviewer #3: Yes

4. Have the authors made all data underlying the findings in their manuscript fully available?

Reviewer #1: Yes

Reviewer #2: (No Response)

Reviewer #3: Yes

5. Is the manuscript presented in an intelligible fashion and written in standard English?

Reviewer #1: Yes

Reviewer #2: Yes

Reviewer #3: Yes

6. Review Comments to the Author

Reviewer #1: Dear Editor,

I reviewed the revised manuscript and found the changes very helpful. Here are my remaining concerns and suggestions for improvement,

I am not satisfied with the response provided regarding the use of logistic regression. The independent variables (predictors) in logistic regression should have minimal or no collinearity. They need to be independent of one another and not to correlate with each other. Therefore, different CEUS parameters cannot be used in the same logistic regression analysis as they are not independent of one another.

I believe a dot plot of actual values of selected CEUS parameters (those that were significantly associated with the study outcomes) for different AKI stages or for those who required RRT versus those who didn’t would be necessary. A visual representation of the differences and overlaps between the groups would be more useful than OR.

CEUS studies were probably performed during different stages of AKI evolution. While a patient may have had their study on day one with minimal rise in serum Cr, another patient may have had it at the peak and a third one while recovering. Reporting the mean difference between the time of CEUS and time of peak serum Cr or nadir of urine output would be helpful in confirming the utility of this tool in prognostication of AKI.

The imaging study of one kidney occurred immediately after injection of the contrast bolus, while the second kidney was images later on. Certain parameters, such as time to peak cannot be obtained for the second kidney. Additionally, since contrast was given a s a bolus and not a continuous infusion, it is expected to have lower image intensity during the study of the second kidney. Please clarify if the data used is only from the first kidney or both and if there were differences between the two.

Minor changes,

Under Study Design and Patients,

Line 84, Adult patients?

Line 84, Change our hospital to the actual name of the hospital

Line 96, move number of patients to results.

Line 98, specify what was measured in urine.

Line 100. Change renal function to “baseline renal function”, since you won’t be using the equation for measuring GFR after AKI.

Line 140-105, what was the frequency of follow up?

Line 106, move AKI staging to where the definition is discussed.

Under methods please indicate how the cause of AKI was determined.

Urine output is used in KDIGO definition of AKI. If that data was recorded and used in categorizing AKI stages, please indicate it under the methods section.

Table 1. I suggest moving CEUS data to a separate table under results.

Reviewer #2: The paper is corrected better.

But I still have a question to ought to be clear, because the renal CEUS is not widely accepted yet. As all sonographic experiences were endured by one radiologist and manual probe supporting, how are they secure the standardization of the examination results?

Reviewer #3: (No Response)

7. PLOS authors have the option to publish the peer review history of their article (what does this mean?). If published, this will include your full peer review and any attached files.

Reviewer #1: Yes: Kambiz Kalantari, MD, MS

Reviewer #2: No

Reviewer #3: Yes: Linda Kelahan

---

## [Author Response · Author response to Decision Letter 1]

21 May 2020

To the editor

We would like to thank the editor and reviewers for their insightful comments. Their comments definitely helped to improve the quality of our manuscript. We have tried our very best to answer the reviewers’ specific points. The reviewer's comments are in back and our answers are in blue. We hope our revisions improve the paper such that it is worthy of publication in PLOS ONE. Thank you. 

Point by point response to reviewers’ comments

[1] Reviewer #1 comments: 

Reviewer #1: Dear Editor,

I reviewed the revised manuscript and found the changes very helpful. Here are my remaining concerns and suggestions for improvement,

1. I am not satisfied with the response provided regarding the use of logistic regression. The independent variables (predictors) in logistic regression should have minimal or no collinearity. They need to be independent of one another and not to correlate with each other. Therefore, different CEUS parameters cannot be used in the same logistic regression analysis as they are not independent of one another.

■ We agree with the reviewer’s comment that CEUS parameters are not independent of each other. The logistic regression analysis was univariate, and included “only one” CEUS parameter for each outcome. Thus, multiple CEUS parameters were not included in a same analysis. 

As the reviewer pointed out in the 1st review, adding other clinical variables would be useful to clearly demonstrate the association between CEUS parameters and outcomes. But as the number of patients and number of outcome events were small, we could not perform a multivariate analysis. 

As our description for the statistical analysis and the results on the logistic regression may confuse the reader, we tried to clarify this by adding the term “univariate” and a sentence “Since CEUS parameters are not completely independent of one another, we included one CEUS parameter for each outcome in the logistic regression analysis.”

2. I believe a dot plot of actual values of selected CEUS parameters (those that were significantly associated with the study outcomes) for different AKI stages or for those who required RRT versus those who didn't would be necessary. A visual representation of the differences and overlaps between the groups would be more useful than OR.

■ We thank the reviewer for this helpful comment. We made a dot plot of actual values of selected CEUS parameters as Figure 4.

3. CEUS studies were probably performed during different stages of AKI evolution. While a patient may have had their study on day one with minimal rise in serum Cr, another patient may have had it at the peak and a third one while recovering. Reporting the mean difference between the time of CEUS and time of peak serum Cr or nadir of urine output would be helpful in confirming the utility of this tool in prognostication of AKI.

■ We agree with the reviewer’s comment. We added the mean, standard deviation, and range of the time of CEUS and time of peak serum Cr in the results, and added this as a limitation of this study. 

“The duration between the time of CEUS and the time of peak serum creatinine was 3.8 ± 3.9 days (range, 0 – 14 days).”

4. The imaging study of one kidney occurred immediately after injection of the contrast bolus, while the second kidney was images later on. Certain parameters, such as time to peak cannot be obtained for the second kidney. Additionally, since contrast was given as a bolus and not a continuous infusion, it is expected to have lower image intensity during the study of the second kidney. Please clarify if the data used is only from the first kidney or both and if there were differences between the two.

■ The contrast agent was injected before examination of each kidney. The right kidney was examined first and, after about 20 min, the same CEUS examinations (including injection of SonoVue) were performed for the left kidney. We added this to clarify how the CEUS was performed. 

“The contrast agent was injected before examination of each kidney. The right kidney was examined first and, after about 20 min, the same CEUS examinations (including injection of SonoVue) were performed for the left kidney.”

We obtained CEUS parameters for both kidneys respectively and additionally calculated the average value of each parameter from both kidneys of each individual. Because the amount of data is too large and difficult to express, only the average value was shown in the paper. A few previous studies conducted in the same way by Wang L et al. [Biomed Res Int. 2015 and J Nephrol 2015]. Although not shown in the paper, we can see that there is no significant difference in the CEUS values of the right and left kidneys, and similar results have been confirmed in some recently published papers [Clin Hemorheol Microcirc 2016;62:229-38, Br J Radiol 2014 Oct;87(1042):20140350. doi: 10.1259/bjr.20140350]. And it has been reported that contrast agent concentrations and GFR are not significantly different between the left and the right kidney [J Vet Med Sci 2016 Feb;78:239-44].

In accordance with this comment, we have added the following sentence to clarify the meaning. 

“The final reported results of CEUS parameters represent the average value of each parameter from both kidneys of each individual.”

Minor changes

Under Study Design and Patients,

1. Line 84, Adult patients?

■ Yes, only adult patients were enrolled. We added the exclusion criteria for age. 

2. Line 84, Change our hospital to the actual name of the hospital

■ We added the actual name of the hospital. 

3. Line 96, move number of patients to results.

■ We moved the number of patients to the result section. 

4. Line 98, specify what was measured in urine.

■ Urine sodium, creatinine, and protein levels were measured. We specified this as the reviewer commented. 

5. Line 100. Change renal function to "baseline renal function", since you won't be using the equation for measuring GFR after AKI.

■ We revised the term as “baseline renal function”.

6. Line 140-105, what was the frequency of follow up?

■ The frequency of follow-up differed between patients. If the patient was cared in the outpatient clinic, the follow-up interval was determined according to the physician’s decision, and ranged from 2 days to 1 week. If the patient was admitted in the hospital, the patient’s status was checked every day. If the patient’s renal function improved to his or her baseline function and the patient did not have CKD, the follow-up was stopped at 3 months post-AKI. Patients with CKD or non-recovery of AKI were followed-up more than 3 months post-AKI, and the follow-up interval was determined according to the physician’s decision. 

This was briefly described in the method section.

7. Line 106, move AKI staging to where the definition is discussed.

■ We revised the sentence as the reviewer commented. 

8. Under methods please indicate how the cause of AKI was determined.

■ The clinical context, history taking, physical examination, and interpretation of blood and urine laboratory findings were used for the differential diagnosis of the cause of AKI. 

This was added in the method section. 

9. Urine output is used in KDIGO definition of AKI. If that data was recorded and used in categorizing AKI stages, please indicate it under the methods section.

■ Not all patients were admitted to the hospital at the diagnosis of AKI. The urine output was calculated for admitted patients. Therefore the serum creatinine criteria was used for the definition and staging for AKI. We clarified such definition in the method section. 

10. Table 1. I suggest moving CEUS data to a separate table under results.

■ We made a separate table for the CEUS data as the reviewer suggested. 

[2] Reviewer #2 comments: 

Reviewer #2: The paper is corrected better.

But I still have a question to ought to be clear, because the renal CEUS is not widely accepted yet. As all sonographic experiences were endured by one radiologist and manual probe supporting, how are they secure the standardization of the examination results?

■ After the injection of the contrast agent, all images and video clips were stored digitally on a hard disk system, and then transferred to a personal computer for further quantitative analyses using advanced US quantification software. As the CEUS parameters are obtained from a time-intensity curve, these parameters are calculated automatically by the software program. Therefore, we can expect that simply measuring these parameters will demonstrate a good agreement among observers, because it is different from characterizing a renal mass.

In general, it can be assumed that renal mass characterization will have greater interobserver variation than simple measurements of parenchymal perfusion. Recently the biggest study cohort of patients with unclear renal mass that were evaluated using CEUS showed a great interobserver agreement between two independent readers [Eur Radiol 2018;28:4542-49]. These findings are in line with several previous studies conducted about this topic.

Many CEUS studies about renal parenchymal perfusion were performed by a single radiologist [Transplant Proc 2009;41:3024-7, Clin Hemorheol Microcirc 2016;62:229-38, Antioxid Redox Signal 2017;27:1397-1411, Abdom Radiol 2018;43:1423-1431]. As previous studies did, our study was also conducted by one researcher.

[3] Reviewer #3 comments: 

 ■ Thank you for reviewing our manuscript.

---

## [Decision Letter · Decision Letter 2]

3 Jun 2020

PONE-D-20-06286R2

A pilot trial to evaluate the clinical usefulness of contrast-enhanced ultrasound in predicting renal outcomes in patients with acute kidney injury

PLOS ONE

Dear Dr. Shin,

Thank you for submitting your manuscript to PLOS ONE. After careful consideration, we feel that it has merit but does not fully meet PLOS ONE’s publication criteria as it currently stands. Therefore, we invite you to submit a revised version of the manuscript that addresses the points raised during the review process.

A reviewer has concern on method.  Please confirm that your method measure medullary blood flow.

We look forward to receiving your revised manuscript.

Kind regards,

Tatsuo Shimosawa, M.D., Ph.D.

Academic Editor

PLOS ONE

Reviewers' comments:

Reviewer's Responses to Questions

**Comments to the Author**

1. If the authors have adequately addressed your comments raised in a previous round of review and you feel that this manuscript is now acceptable for publication, you may indicate that here to bypass the “Comments to the Author” section, enter your conflict of interest statement in the “Confidential to Editor” section, and submit your "Accept" recommendation.

Reviewer #1: All comments have been addressed

Reviewer #2: All comments have been addressed

2. Is the manuscript technically sound, and do the data support the conclusions?

Reviewer #1: Yes

Reviewer #2: Partly

3. Has the statistical analysis been performed appropriately and rigorously? 

Reviewer #1: Yes

Reviewer #2: Yes

4. Have the authors made all data underlying the findings in their manuscript fully available?

Reviewer #1: Yes

Reviewer #2: Yes

5. Is the manuscript presented in an intelligible fashion and written in standard English?

Reviewer #1: Yes

Reviewer #2: Yes

6. Review Comments to the Author

Reviewer #1: Thank you for making all the suggested changes. The manuscript reads very well. I just wanted to mention that in my experience, the CEUS data for medulla differ significantly from that of cortex and I was surprised to see them identical in your study.

Reviewer #2: Thank you for the response. I still have a technical question. Physiologically and anatomically it is well known that the blood flow and vessels of medulla are much less than that of cortex. It feels strange the blood flow of medulla looks similar to cortex in Figure 3. But it could be limitations for the quantitative CEUS as the authors discussed.

7. PLOS authors have the option to publish the peer review history of their article (what does this mean?). If published, this will include your full peer review and any attached files.

Reviewer #1: Yes: Kambiz Kalantari, MD, MS

Reviewer #2: No

---

## [Author Response · Author response to Decision Letter 2]

5 Jun 2020

To the editor

We would like to thank the editor and reviewers for their insightful comments. Their comments definitely helped to improve the quality of our manuscript. We have tried our very best to answer the reviewers’ specific points. The reviewer's comments are in blue and our answers are in black. We hope our revisions improve the paper such that it is worthy of publication in PLOS ONE. Thank you. 

Point by point response to reviewers’ comments

[1] Reviewer #1 comments: 

Thank you for making all the suggested changes. The manuscript reads very well. I just wanted to mention that in my experience, the CEUS data for medulla differ significantly from that of cortex and I was surprised to see them identical in your study.

We thank the reviewer for this informative comment and agree with you. Unlike medulla, which normally receives less flow than cortex, we speculate that a medullary blood congestion and consequent slowing of the blood flow occurred in our patients with AKI. This medullary congestion is a hallmark of ischemic AKI in both experimental models (Kidney Int 1990; 37: 1240-1247, Kidney Int 1990; 38: 432-439, Kidney Int 1991; 40: 625-631, Lab Invest 1991; 65: 566-576) and in specimens obtained from biopsy or at autopsy (Kidney Int 1984; 26: 283-293). Currently, there is no CEUS study for medullary perfusion in humans with AKI. A recent animal study showed that medullary PI and AUC were about one-third of that of cortex in healthy dogs, on the other hand, medullary PI and AUC were almost the same as cortex values in dogs with AKI (J Small Anim Pract. 2019 Aug;60(8):471-476). The increase in medullary PI and AUC seen in dogs with AKI, which suggests increased medullary congestion, is similar to our results. However, our study had patients with mild AKI, whereas few papers on human CEUS had severe illness. Further studies with a larger sample size are needed for validation.

We added the following sentence about the medullary CEUS findings in discussion:

“In this study, the CEUS-driven TIC parameters of the medulla were not reduced compared to those of cortex. This was unexpected, as the renal medulla receives lower blood flow than the cortex (Seminars in Nephrology, Vol 39, No 6, November 2019, 520−529). We speculate that a medullary blood congestion and consequent slowing of the blood flow occurred in our patients with AKI. This medullary congestion is a hallmark of ischemic AKI in both experimental models (Kidney Int 1990; 37: 1240-1247, Kidney Int 1990; 38: 432-439, Kidney Int 1991; 40: 625-631, Lab Invest 1991; 65: 566-576) and in specimens obtained from biopsy or at autopsy (Kidney Int 1984; 26: 283-293). Currently, there is no CEUS study for medullary perfusion in humans with AKI. A recent animal study showed that medullary PI and AUC were about one-third of that of cortex in healthy dogs, on the other hand, medullary PI and AUC were almost the same as cortex values in dogs with AKI (J Small Anim Pract. 2019 Aug;60(8):471-476). The increase in medullary PI and AUC seen in dogs with AKI, which suggests increased medullary congestion, is similar to our results. Further studies with a larger sample size are needed for validation of medullary CEUS findings in patients with AKI.”

[2] Reviewer #2 comments: 

Thank you for the response. I still have a technical question. Physiologically and anatomically it is well known that the blood flow and vessels of medulla are much less than that of cortex. It feels strange the blood flow of medulla looks similar to cortex in Figure 3. But it could be limitations for the quantitative CEUS as the authors discussed.

We thank the reviewer for this informative comment and agree with you. Unlike medulla, which normally receives less flow than cortex, we speculate that a medullary blood congestion and consequent slowing of the blood flow occurred in our patients with AKI. This medullary congestion is a hallmark of ischemic AKI in both experimental models (Kidney Int 1990; 37: 1240-1247, Kidney Int 1990; 38: 432-439, Kidney Int 1991; 40: 625-631, Lab Invest 1991; 65: 566-576) and in specimens obtained from biopsy or at autopsy (Kidney Int 1984; 26: 283-293). Currently, there is no CEUS study for medullary perfusion in humans with AKI. A recent animal study showed that medullary PI and AUC were about one-third of that of cortex in healthy dogs, on the other hand, medullary PI and AUC were almost the same as cortex values in dogs with AKI (J Small Anim Pract. 2019 Aug;60(8):471-476). The increase in medullary PI and AUC seen in dogs with AKI, which suggests increased medullary congestion, is similar to our results. However, our study had patients with mild AKI, whereas few papers on human CEUS had severe illness. Further studies with a larger sample size are needed for validation.

We added the discussion about the medullary CEUS findings as above.

---

## [Decision Letter · Decision Letter 3]

10 Jun 2020

A pilot trial to evaluate the clinical usefulness of contrast-enhanced ultrasound in predicting renal outcomes in patients with acute kidney injury

PONE-D-20-06286R3

Dear Dr. Shin,

We’re pleased to inform you that your manuscript has been judged scientifically suitable for publication and will be formally accepted for publication once it meets all outstanding technical requirements.

Kind regards,

Tatsuo Shimosawa, M.D., Ph.D.

Academic Editor

PLOS ONE

Additional Editor Comments (optional):

Reviewers' comments:

Reviewer's Responses to Questions

**Comments to the Author**

1. If the authors have adequately addressed your comments raised in a previous round of review and you feel that this manuscript is now acceptable for publication, you may indicate that here to bypass the “Comments to the Author” section, enter your conflict of interest statement in the “Confidential to Editor” section, and submit your "Accept" recommendation.

Reviewer #2: All comments have been addressed

2. Is the manuscript technically sound, and do the data support the conclusions?

Reviewer #2: Yes

3. Has the statistical analysis been performed appropriately and rigorously? 

Reviewer #2: Yes

4. Have the authors made all data underlying the findings in their manuscript fully available?

Reviewer #2: Yes

5. Is the manuscript presented in an intelligible fashion and written in standard English?

Reviewer #2: Yes

6. Review Comments to the Author

Reviewer #2: Than you for your response. I agree the comment. And I believe that this report is important for the future clinical relevant of acute renal injury.

7. PLOS authors have the option to publish the peer review history of their article (what does this mean?). If published, this will include your full peer review and any attached files.

Reviewer #2: No

---

## [Editor Report · Acceptance letter]

15 Jun 2020

PONE-D-20-06286R3 

A pilot trial to evaluate the clinical usefulness of contrast-enhanced ultrasound in predicting renal outcomes in patients with acute kidney injury 

Dear Dr. Shin:

I'm pleased to inform you that your manuscript has been deemed suitable for publication in PLOS ONE. Congratulations! Your manuscript is now with our production department. 

Kind regards, 

on behalf of

Prof. Tatsuo Shimosawa 

Academic Editor

PLOS ONE